# Phosphoric acid-catalyzed atroposelective construction of axially chiral arylpyrroles

Lei Zhang[1], Shao-Hua Xiang [1,2], Jun (Joelle) Wang [1], Jian Xiao[3], Jun-Qi Wang[4] & Bin Tan [1]

Axially chiral arylpyrroles are key components of pharmaceuticals and natural products as well as chiral catalysts and ligands for asymmetric transformations. However, the catalytic enantioselective construction of optically active arylpyrroles remains a formidable challenge. Here we disclose a highly efficient strategy to access enantioenriched axially chiral arylpyrroles by means of organocatalytic atroposelective desymmetrization and kinetic resolution. Depending on the remote control of chiral catalyst, the arylpyrroles were obtained in high yields and excellent enantioselectivities under mild reaction conditions. This strategy tolerates a wide range of functional groups, providing a facile avenue to approach axially chiral arylpyrroles from simple and readily available starting materials. Selected arylpyrrole products proved to be efficient chiral ligands in asymmetric catalysis and also important precursors for further synthetic transformations into highly functionalized pyrroles with potential bioactivity, especially the axially chiral fully substituted arylpyrroles.

[1] Department of Chemistry and Shenzhen Grubbs Institute, Southern University of Science and Technology, Shenzhen 518055, China. [2] Academy for Advanced Interdisciplinary Studies, Southern University of Science and Technology, Shenzhen 518055, China. [3] College of Chemistry and Pharmaceutical Sciences, Qingdao Agricultural University, Qingdao 266109, China. [4] Department of Biology, Southern University of Science and Technology, Shenzhen 518055, China. Correspondence and requests for materials should be addressed to J.W. (email: wang.j@sustc.edu.cn) or to B.T. (email: tanb@sustc.edu.cn)

Axially chiral arylpyrroles constitute the core skeletons of a wide range of natural products[1–3] and pharmaceutical agents[4–6]. Owing to their intrinsic structure characteristics, they proved to possess widespread applications in organic synthesis[7–15]. For example, arylpyrroles scaffolds occur in plenty of chiral phosphine ligands applied in numerous transition metal-mediated reactions (Fig. 1)[7–9]. More importantly, the optically pure arylpyrrole derivatives have been extensively used in current asymmetric reactions[10–14] as chiral resolving agents[10,11], chiral ligands[12,13] as well as chiral catalysts[14]. Nonetheless, over the past decade, conventional approaches to access these optically active arylpyrroles were by optical resolution of the racemates using chiral resolution agents or chiral column chromatography, which not only required the stoichiometric amounts of chiral reagent, but also were limited by the substrate scope[10,15,16]. While the first catalytic asymmetric Paal–Knorr reaction was established to access highly enantioenriched axially chiral arylpyrroles by our group recently[17], complicated catalytic system and narrow substrate range restricted its application. Therefore, the atroposelective construction of axially chiral arylpyrroles remains comparatively unexplored and the development of facile and direct route is highly appealing.

Recently, Bencivenni's group reported a pioneering study on the remote control of the axial chirality of atropisomeric succinimides through vinylogous Michael addition reaction via aminocatalytic desymmetrization of arylmaleimides (Fig. 2a)[18–20]. The key feature of this method relies on the recognition of the catalyst with regards to the maleimide's symmetry plane. The hydrogen bonding between the oxygen of the carbonyl group of the substrate and the protonated quinuclidine moiety of the catalyst is crucial for the excellent stereoselectivity. Inspired by this elegant work involving the remote control of the axial chirality, as well as our ongoing interest in chiral phosphoric acid (CPA) catalytic atroposelective synthesis of enantioenriched axially chiral structures[21,22], we envisage that the readily available 2,5-disubstituted arylpyrroles should be suitable prochiral substrates to afford the expected axially chiral structures by enantioselective desymmetrization[23–26] or kinetic resolution[27–32] under the strategy of remote stereocontrol by organocatalysts (Fig. 2b)[33–36]. However, the major challenges in this scenario would be: (1) the choice of appropriate electrophilic reagents could react efficiently with the arylpyrroles. Meanwhile, the electrophiles would interact with the organocatalyst in a reasonable spatial configuration to enhance the reactivity and offer an ideal chiral environment for the control of the atroposelectivity; (2) the careful selection of compatible chiral organocatalyst to effectively induce the axial chirality since the distance between the chiral axis and the catalyst activation site is relatively long; (3) lacking an available proton donor group on the arylpyrroles for hydrogen bonding effect, the interaction between arylpyrrole substrates and the chiral catalyst could not occur effectively[37–44]. Herein, we present our strategy to overcome the abovementioned challenges and successfully construct the highly enantioenriched axially chiral arylpyrroles by means of the remote control of the axial chirality. Two complementary approaches, namely desymmetrization/kinetic resolution, are devised with CPA as the catalyst to give the axially chiral arylpyrroles with highly structural diversity and excellent enantiocontrol. These optical active axially chiral arylpyrroles are able to sever as chiral building blocks for rapid transformations to functionalized pyrroles with potential bioactivity and ligands for asymmetric catalytic reactions.

## Results

**Reaction condition optimization.** To validate the feasibility of the hypothesis, the reaction of 1-(2-(*tert*-butyl)phenyl)-2,5-dimethyl-1*H*-pyrrole **1a** and diethyl ketomalonate **2a** were conducted with 10 mol% SPINOL-derived CPA (*S*)-**C1** in toluene at room temperature. Encouragingly, the reaction proceeded smoothly to give the desired axially chiral arylpyrrole **3a** in 74% yield (Table 1, entry 1). Despite only 22% ee, this result clearly demonstrated that the organocatalytic remote control of the chiral axis of arylpyrroles by desymmetrization strategy was feasible. To improve reaction outcomes, we turned our attention to explore the effect of the catalysts. As shown in Table 1, all the tested CPAs were capable of catalyzing this reaction to give **3a** in reasonable to excellent yields (Table 1, entries 2–8). These results also evidenced that CPAs containing bulky substituents could significantly enhance the stereocontrol of the reaction and H$_8$-BINOL-derived (*S*)-**C8** with 2,4,6-triisopropylphenyl group on the 3,3′-position was superior to other catalysts to give the desired product **3a** in 93% yield with 90% ee. After that, various solvents were evaluated to prove the cyclohexane as the optimal reaction solvent (Table 1, entries 8–12). Lower reaction temperature resulted in negligible effect in enantioselectivity, accompanied by

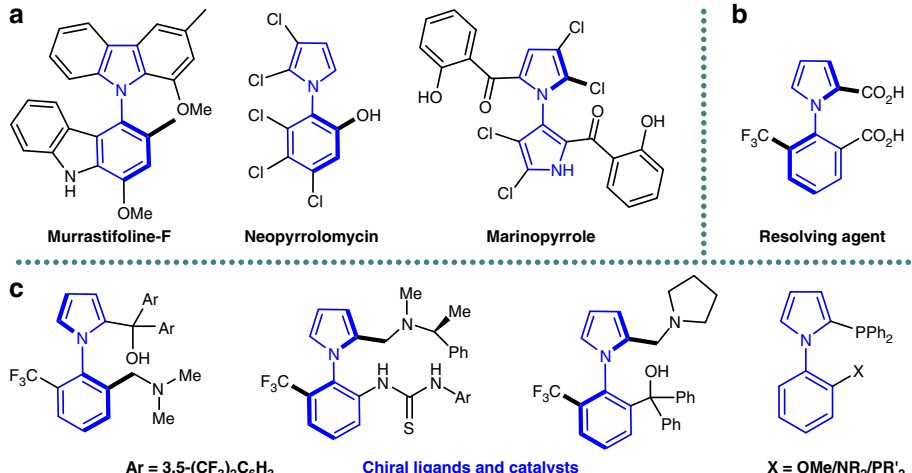

**Fig. 1** Representative molecules containing axially chiral arylpyrrole frameworks. **a** Bioactive natural products. **b** Resolving agent. **c** Chiral ligands and catalysts

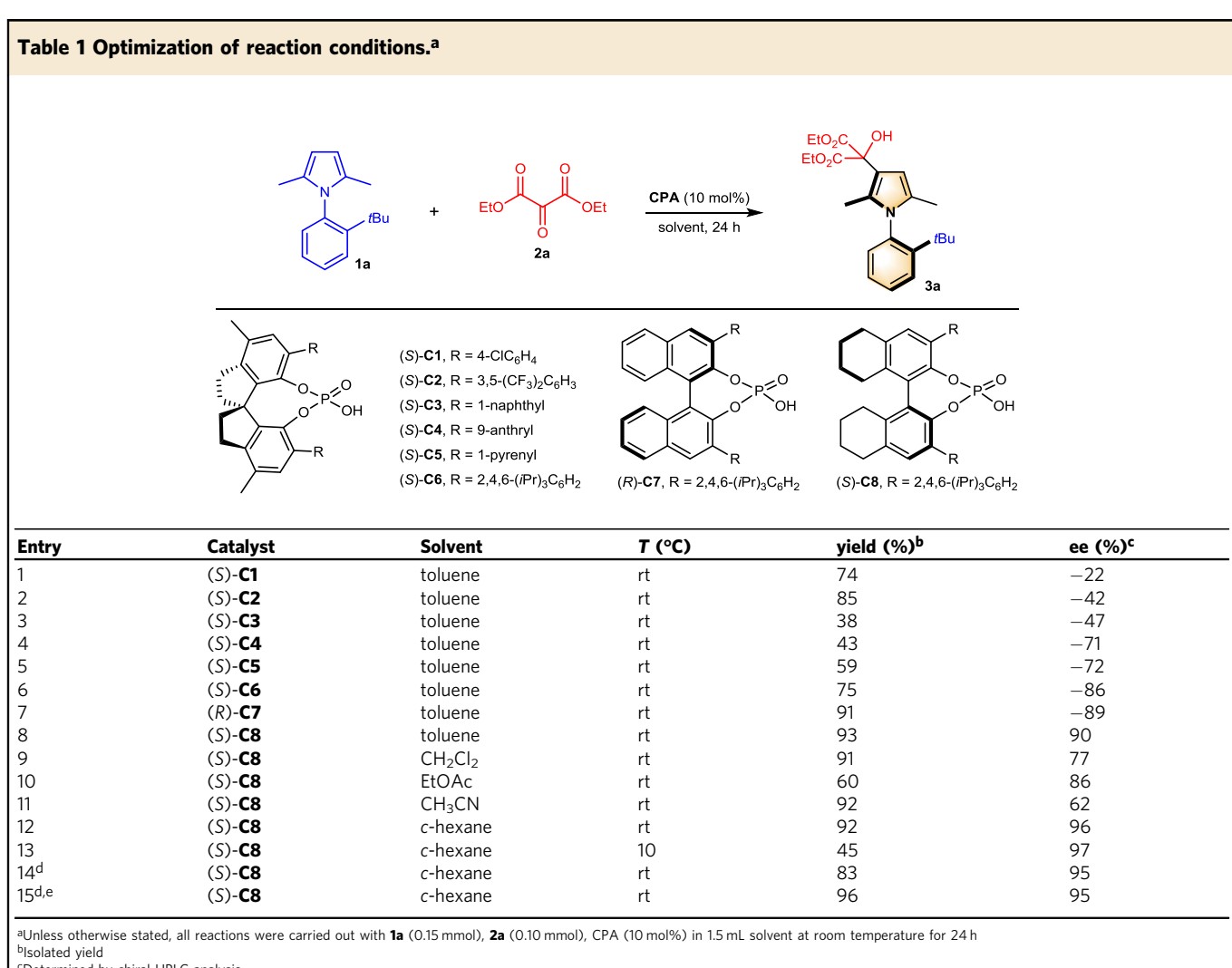

**Fig. 2** Background and project synopsis. **a** Aminocatalytic enantioselective synthesis of atropisomeric succinimides via remote control strategy (Bencivenni's work). **b** Our strategy for the remote enantiocontrol of axially chiral arylpyrroles. Black circle, sterically bulky substituent

### Table 1 Optimization of reaction conditions.[a]

| Entry | Catalyst | Solvent | T (°C) | yield (%)[b] | ee (%)[c] |
|---|---|---|---|---|---|
| 1 | (S)-**C1** | toluene | rt | 74 | −22 |
| 2 | (S)-**C2** | toluene | rt | 85 | −42 |
| 3 | (S)-**C3** | toluene | rt | 38 | −47 |
| 4 | (S)-**C4** | toluene | rt | 43 | −71 |
| 5 | (S)-**C5** | toluene | rt | 59 | −72 |
| 6 | (S)-**C6** | toluene | rt | 75 | −86 |
| 7 | (R)-**C7** | toluene | rt | 91 | −89 |
| 8 | (S)-**C8** | toluene | rt | 93 | 90 |
| 9 | (S)-**C8** | CH₂Cl₂ | rt | 91 | 77 |
| 10 | (S)-**C8** | EtOAc | rt | 60 | 86 |
| 11 | (S)-**C8** | CH₃CN | rt | 92 | 62 |
| 12 | (S)-**C8** | c-hexane | rt | 92 | 96 |
| 13 | (S)-**C8** | c-hexane | 10 | 45 | 97 |
| 14[d] | (S)-**C8** | c-hexane | rt | 83 | 95 |
| 15[d,e] | (S)-**C8** | c-hexane | rt | 96 | 95 |

[a]Unless otherwise stated, all reactions were carried out with **1a** (0.15 mmol), **2a** (0.10 mmol), CPA (10 mol%) in 1.5 mL solvent at room temperature for 24 h
[b]Isolated yield
[c]Determined by chiral HPLC analysis
[d](S)-**C8** (5 mol%) was used
[e]Reaction was allowed to stir at room temperature for 36 h

**Table 2 Substrate scope with respect to symmetric prochiral arylpyrroles.[a]**

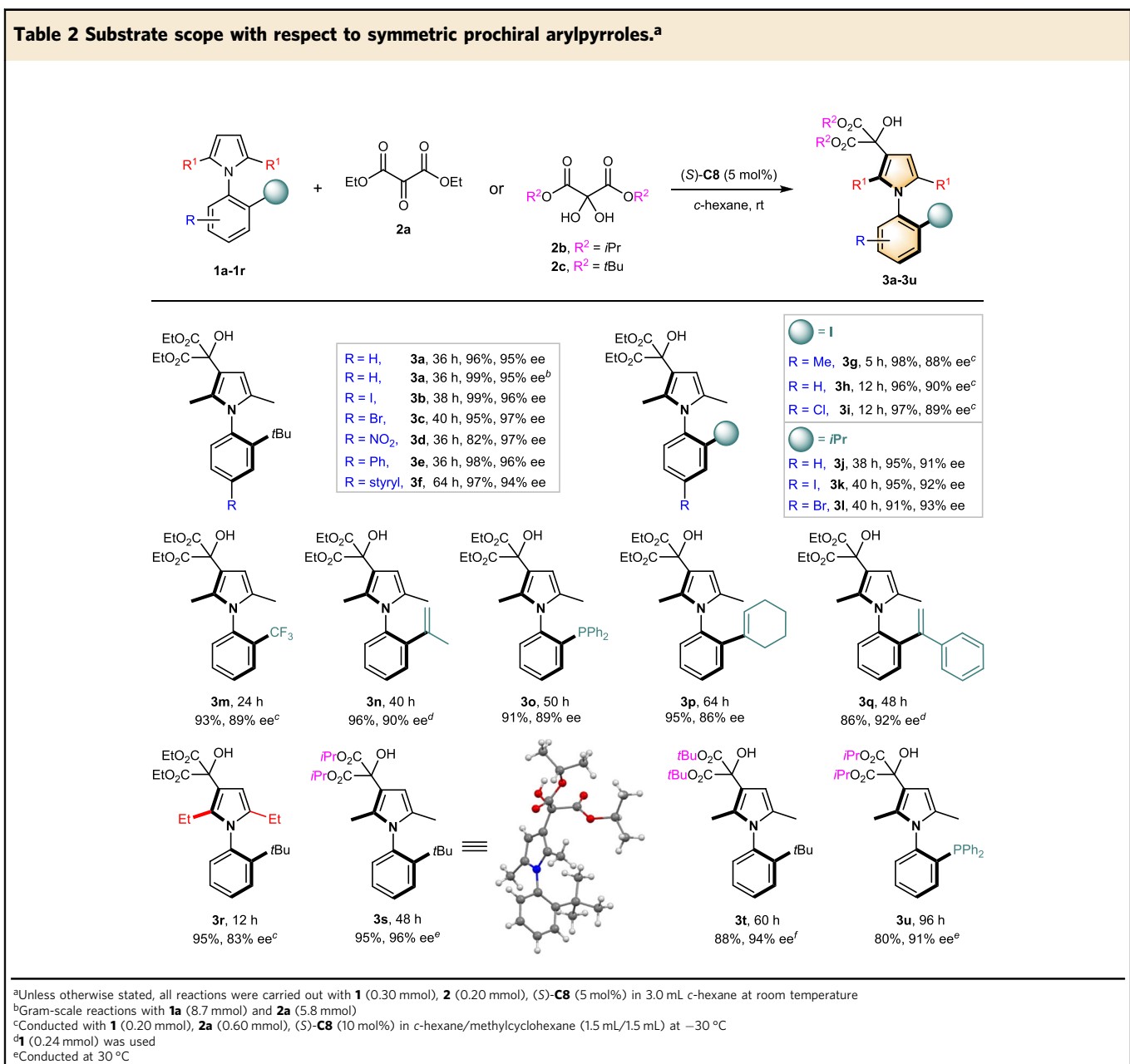

[a]Unless otherwise stated, all reactions were carried out with **1** (0.30 mmol), **2** (0.20 mmol), (S)-**C8** (5 mol%) in 3.0 mL c-hexane at room temperature
[b]Gram-scale reactions with **1a** (8.7 mmol) and **2a** (5.8 mmol)
[c]Conducted with **1** (0.20 mmol), **2a** (0.60 mmol), (S)-**C8** (10 mol%) in c-hexane/methylcyclohexane (1.5 mL/1.5 mL) at −30 °C
[d]**1** (0.24 mmol) was used
[e]Conducted at 30 °C
[f]Conducted at 50 °C. Styryl, phenyl vinyl

the loss of chemical yield (Table 1, entry 13). Further optimization revealed that 5 mol% catalyst loading could uphold the enantioselectivity effectively with slight increase of chemical yield when the reaction time was extended to 36 h (Table 1, entries 14 and 15). Based on the above results (See Supplementary Tables 1–4), the optimized conditions were concluded as follows: treatment of **2a** with 1.5 equivalent of **1a** in cyclohexane at room temperature in the presence of 5 mol% of (S)-**C8** and the reaction could give the desired axially chiral arylpyrrole **3a** in 96% isolated yield with 95% ee (Table 1, entry 15).

**Substrate scope.** After establishing the optimal reaction conditions, we set out to explore the substrate generality of this transformation. Firstly, the substrate scope with respect to the symmetrical arylpyrroles was investigated (**1a–1r**). Most reactions could be completed within 64 h to give the desired axially chiral

arylpyrroles (Table 2, **3a–3r**) in high yields (82–99%) with excellent enantioselectivities (83–97% ee). It is noteworthy to point out that the ortho group on the phenyl ring is not only restricted to tbutyl group, iodo, isopropyl, trifluoromethyl, diphenyl phosphine, 2-isopropenyl, styrene as well as cyclohexene groups were also applicable to give the expected products with good to excellent enantiocontrol (**3g–3q**). Subsequently, symmetrical arylpyrroles **1r** bearing diethyl group could afford **3r** in excellent yield and good enantioselectivity. Moreover, varying the ester moiety of ketomalonate could also result in excellent enantioselectivities (**3s–3u**). Notably, **1o** with a diphenyl phosphine substituent was converted to the corresponding product **3o** and **3u** as potential chiral phosphine ligands. Besides, the halide substituents in the obtained arylpyrroles provided opportunities for the downstream coupling reactions to set up a compound library by routine transition metal-catalyzed reactions.

Furthermore, the absolute configurations of the axially chiral products were assigned to be (*aR*) by X-ray crystallographic analysis of **3s** and those of the other products were assigned by analogy (see Supplementary Figure 1). To verify the practicality of such method, a gram-scale reaction was carried out for the synthesis of **3a** under the optimal reaction conditions. As displayed in Table 2, no significant variation was detected in terms of the chemical yield and stereoselectivity as compared to the small scale reaction.

Encouraged by the above results about the construction of axially chiral arylpyrroles via atroposelective desymmetrization of symmetric substrates, we continued to explore the feasibility of this reaction with asymmetric racemic arylpyrroles by using kinetic resolution to further expand the substrate scope. Plenty of asymmetric arylpyrrole racemates with aromatic substituents on the pyrrole ring were then prepared and subjected to the standard conditions. The results were summarized in Table 3 and all the reactions underwent smoothly to give good to high selectivity factor (*S* = 32−69, **5a–5l**). The corresponding products were obtained in 87−94% ee with 42−47% isolated yields regardless of the steric and electronic properties of the substituents. Notably, substrates possessing a naphthyl or thienyl group (**5j–5l**) also worked efficiently to afford high selectivity, respectively. Moreover, replacement of aromatic substitutions with alkyl group gave rise to unobvious effect on the reaction outcomes (**5m–5n**). Our method thus represents one of the most straightforward syntheses of axially chiral arylpyrrole and its analogs. The absolute configuration of **5a** was attributed to be (*aS*) by X-ray crystallographic analysis and those of the other products were assigned by analogy (see Supplementary Figure 2).

**Versatile synthetic transformations**. To demonstrate the synthetic utility of the obtained axially chiral arylpyrroles, a series of chemical transformations were then conducted (Fig. 3a). Firstly, the free OH group of the axially chiral arylpyrrole **3a** could be easily protected by a benzyl group to yield compound **6**. Next, the treatment of compound **3a** with thiourea or methylamine gave the corresponding diamide **7** or thiobarbituric structure **8** in almost quantitative yield without any erosion of enantioselectivity[45,46]. While highly functionalized pyrroles show a wide spectrum of bioactivities, very few methods are available to access these frameworks[47], particularly penta-substituted axially chiral arylpyrroles. Gratifying, the fully substituted axially chiral pyrroles **9**, **10**, **11**, and **12** could be generated from **3a** by classic Mannich and Vilsmeier–Haack reaction, respectively in moderate yields with highly preserved chiral integrity[48]. Subsequently, bromination of **3a** with NBS gave axially chiral arylpyrrole **13** in 80% yield, which could be employed as a precursor to prepare fully substituted axially chiral pyrroles by the following coupling reactions. Furthermore, the treatment of **3a** with 4-methylbenzenesulfonylisocyanate provided compound **14** without loss of stereochemical integrity. Noteworthy, arylpyrrole (*R*)-**3a** could be readily oxidized to dialdehyde **15** with cerium(IV) ammonium nitrate as the oxidative reagent[49]. Moreover, the obtained product **3h** possessing an iodo substituent was an applicable reactant for classic transition metal-mediated coupling reactions (Fig. 3b). For instance, the Sonogashira reaction proceeded smoothly to furnish the desired product **16** with a synthetic useful alkynyl group in 82% yield, while compound **17** was produced efficiently by Suzuki coupling with organoboronic acid in the presence of palladium catalyst. Apart from that, the verified axially chiral phosphine ligand **3o** could also be synthesized from **3h** via rapid C−P bond formation in 66% yield. Notably, no ee erosion was detected for all these reactions.

**Configurational stability test and catalytic applications**. The investigation for the configurational stability of the product was conducted by heating a solution of **3a** in different solvent (*i*PrOH, DCE, and toluene) at up to 150 °C for 24 h. Deteriorations of stereochemical integrity were negligible even under the conditions which the substrate began to decompose. Similar experimental results were obtained with **3o** and **5c** as the test objects (see Supplementary Table S6–S8). Therefore, this kind of axially chiral compounds displayed a high-rotation energy and may have potential applications as organocatalysts/ligands. Finally, we evaluated the applicability of the resulted axially chiral arylpyrrole in asymmetric catalysis. Compound **3o** (99% ee, after semi-preparative high-performance liquid chromatography (HPLC) enantioseparation) was then selected as the ligand for the palladium catalyzed allylic alkylation[50]. Gratifyingly, the reaction of racemic **18** and malonate **19** proceeded effectively with 2 mol% of palladium catalyst and 4 mol% of **3o** to give the desired product **20** in 95% yield with 97% ee. Aside from **3o**, compound **3u** (99% ee, recrystallization with Et₂O/hexane) could also work well in this type of reaction with indole as the nucleophile, indicating that the resulted highly enantioenriched axially chiral arylpyrrole is capable of inducing the chirality in asymmetric synthesis (Fig. 4a). Further studies with regard to the application of axially chiral arylpyrroles as catalysts or ligands for asymmetric reactions are currently under active investigations.

**Plausible mechanism**. A monofunctional activation mode was proposed in Fig. 4b for this asymmetric transformation based on the experimental results and the reports from Terada[51] and Rueping[52]. The hydrogen bonding between ketomalonate and CPA was the pivotal interaction to form the chiral pocket[37] for the induction of chirality. The second carbonyl of the ketomalonate is essential to fix the whole system in a rigid configuration. Besides, such a pathway basically coincides with the plausible mechanism provided by List and Liu et al. for the addition of indolizines and *N*-methylpyrroles to enones and imines[38–40].

## Discussion

In conclusion, we have successfully developed a highly efficient and practical approach for the atroposelective synthesis of axially chiral arylpyrrole derivatives through two complementary asymmetric transformations (desymmetrization/kinetic resolution). Excellent yields and enantioselectivities were obtained with CPA as the chiral catalyst. The key feature of this approach is the achievement of axial chirality via remote control manner, allowing the chiral catalyst to transfer its stereochemical properties along the C−N axis of arylpyrrole. Moreover, highly enantioenriched arylpyrroles proved to be efficient chiral ligands in asymmetric catalysis and versatile building blocks to access other useful axially chiral molecules. Diversified synthetic transformations demonstrated the utilities of this approach, yielding a variety of functionalized axially chiral arylpyrroles, especially the axially chiral penta-substituted arylpyrroles.

## Methods

**General information**. Chemicals were purchased from commercial suppliers and used without further purification unless otherwise stated. CPA was purchased from Daicel Chiral Technologies (China). Analytical thin layer chromatography (TLC) was performed on precoated silica gel 60 GF254 plates. Flash column chromatography was performed using Tsingdao silica gel (60, particle size 0.040–0.063 mm). Visualization on TLC was achieved by use of UV light (254 nm) or iodine. Nuclear magnetic resonance (NMR) spectra were recorded on a Bruker DPX 400 spectrometer at 400/500 MHz for ¹H NMR, 100/125 MHz for ¹³C NMR and 376 MHz for ¹⁹F NMR in CDCl₃, DMSO-d₆ with tetramethylsilane (TMS) as internal standard. The chemical shifts are expressed in ppm and coupling constants are given in Hz. Data for ¹H NMR are recorded as follows: chemical shift (δ, ppm), multiplicity (s

**Table 3 Substrate scope with respect to asymmetric racemic arylpyrroles.**[a]

| Entry | t (d) | Yield of 4 (%)[b] | ee of 4 (%)[c] | Yield of 5 (%)[b] | ee of 5 (%)[c] | Conv. (%) | S[d] |
|---|---|---|---|---|---|---|---|
| 1 | 4.0 | 52 (**4a**) | 72 | 43 (**5a**) | 91 | 44 | 46 |
| 2 | 3.0 | 51 (**4b**) | 80 | 45 (**5b**) | 89 | 47 | 42 |
| 3 | 3.5 | 52 (**4c**) | 76 | 44 (**5c**) | 90 | 46 | 44 |
| 4 | 4.0 | 49 (**4d**) | 89 | 46 (**5d**) | 91 | 49 | 63 |
| 5 | 3.5 | 51 (**4e**) | 73 | 44 (**5e**) | 91 | 45 | 46 |
| 6 | 3.5 | 52 (**4f**) | 76 | 45 (**5f**) | 88 | 46 | 36 |
| 7 | 6.0 | 50 (**4g**) | 77 | 45 (**5g**) | 92 | 46 | 56 |
| 8 | 3.5 | 48 (**4h**) | 81 | 47 (**5h**) | 89 | 48 | 43 |
| 9 | 3.5 | 52 (**4i**) | 72 | 43 (**5i**) | 90 | 44 | 41 |
| 10 | 3.5 | 51 (**4j**) | 81 | 46 (**5j**) | 91 | 47 | 53 |
| 11 | 3.5 | 51 (**4k**) | 75 | 45 (**5k**) | 87 | 46 | 32 |
| 12 | 4.0 | 54 (**4l**) | 71 | 42 (**5l**) | 94 | 43 | 69 |
| 13 | 3.5 | 50 (**4m**) | 82 | 45 (**5m**) | 91 | 47 | 54 |
| 14 | 3.5 | 46 (**4n**) | 85 | 47 (**5n**) | 90 | 49 | 51 |

[a]All reactions were carried out with (rac)-**4** (0.40 mmol), **2a** (0.20 mmol), (S)-**C8** (10 mol%) in cyclohexane (4.8 mL) at 30 °C
[b]Isolated yield
[c]Determined by chiral stationary phase HPLC analysis
[d]The selectivity factor was calculated as $S = \ln[(1 - C)(1 - \text{ee}(\mathbf{4}))]/\ln[(1 - C)(1 + \text{ee}(\mathbf{4}))]$, $C = \text{ee}(\mathbf{4})/(\text{ee}(\mathbf{5}) + \text{ee}(\mathbf{4}))$

= singlet; d = doublet; t = triplet; q = quartet; p = pentet; m = multiplet; br = broad), coupling constant (Hz), integration. Data for [13]C NMR are reported in terms of chemical shift (δ, ppm). Mass spectrometric data were obtained using Bruker Apex IV RTMS. The enantiomeric excess values were determined by chiral HPLC with an Agilent 1200 LC instrument and CHIRALPAK and CHIRALCEL columns. High resolution mass spectroscopy analyses were performed at a Bruker Daltonics. Inc mass instrument (electrospray ionization (ESI)), Thermo Scientific. Q-Exactive (heated ESI (HESI)), and Thermo Scientific. Orbitrap Fusion (HESI).

**General procedure for the synthesis of racemic 3**. An oven-dried 10 mL of Schlenk tube was charged with arylpyrroles **1** (0.15 mmol), 1 mL of CH₂Cl₂ and diphenyl phosphate (0.01 mmol) at ambient temperature. Then, ketomalonate **2** (0.10 mmol) was added to the above solution and the mixture was stirred until the starting material was completely consumed. The mixture was concentrated under reduced pressure and purified by flash column chromatography (ethyl acetate/petroleum ether) to afford the corresponding racemic product **3**.

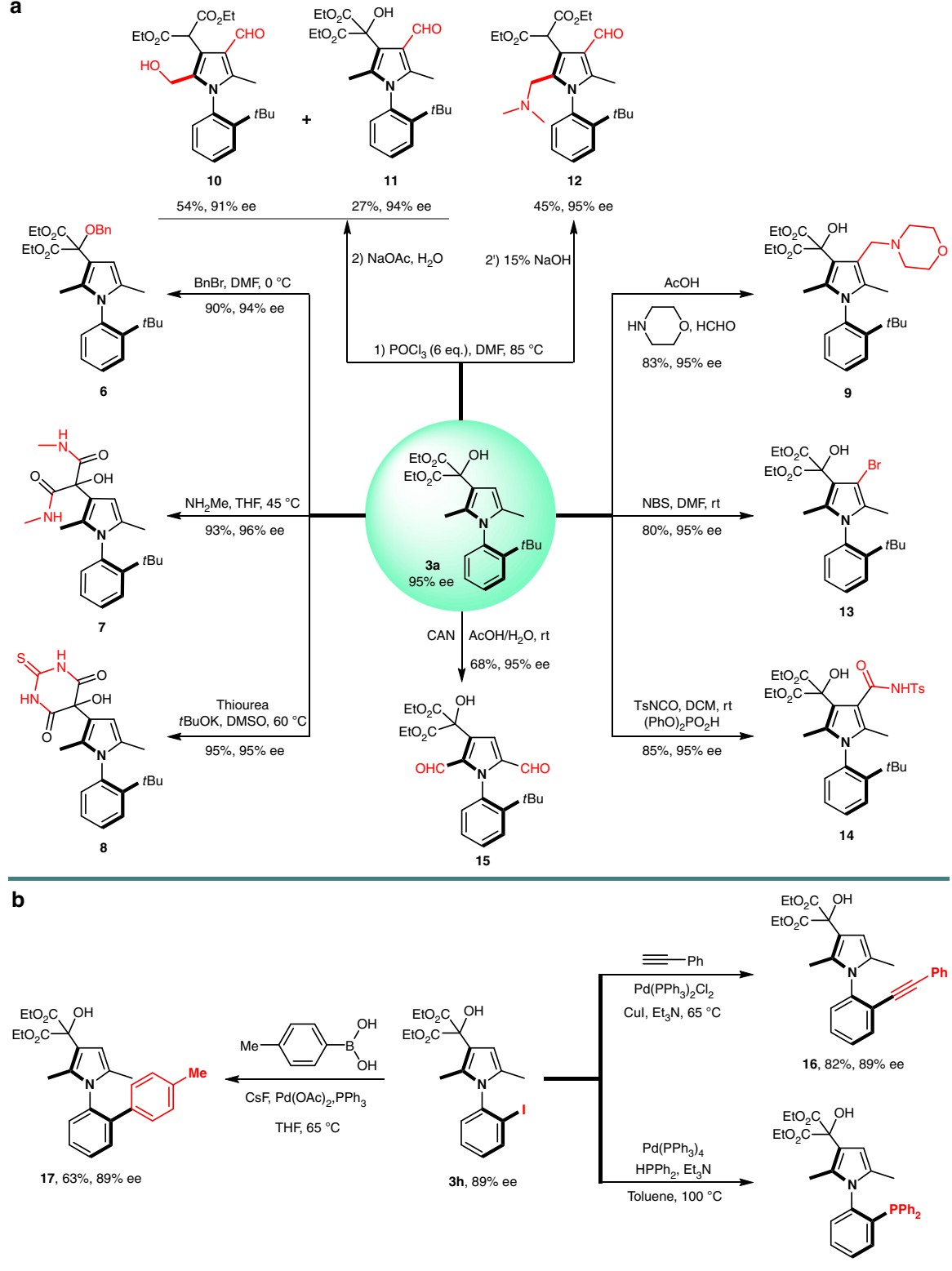

**Fig. 3** Versatile synthetic transformations. **a** Synthetic transformations of compound **3a**. Ts, *p*-toluenesulfonyl. **b** Synthetic transformations of compound **3h**

**General procedure for the synthesis of (*R*)-3.** An oven-dried 10 mL of Schlenk tube was charged with arylpyrroles **1** (0.30 mmol), (*S*)-**C8** (0.01 mmol), 2.0 mL of dry cyclohexane, and the mixture was stirred at ambient temperature for 10 min. A solution of ketomalonate **2a** (0.20 mmol) in dry cyclohexane (1.0 mL) was added dropwise to the above solution and the mixture was stirred until the starting material was completely consumed. Then the mixture was concentrated under reduced pressure and purified by flash chromatography eluted with PE/EA (10/1 to 5/1) to afford the corresponding axially chiral arylpyrrole products (*R*)-**3**.

For **3g–3i, m, 3r**, the reaction conditions are as follows: an oven-dried 10 mL of Schlenk tube was charged with arylpyrroles **1** (0.30 mmol), (*S*)-**C8** (0.02 mmol) and 1.5 mL of mixed solvent (cyclohexane/methylcyclohexane = 1/1). After the

**Fig. 4** Applications in asymmetric catalysis and plausible mechanism. **a** Asymmetric catalysis applications. **b** Proposed reaction mechanism

mixture was stirred at −30 °C for 30 min, a solution of ketomalonate **2a** (0.20 mmol) in 1.5 mL of mixed solvent (0.75 mL cyclohexane/0.75 mL methylcyclohexane) was added dropwise to the above solution and the mixture was stirred until the starting material was completely consumed. Then the mixture was concentrated under reduced pressure and purified by flash chromatography eluted with PE/EA (10/1 to 5/1) to afford the corresponding axially chiral arylpyrrole products.

**General procedure for the synthesis of racemic 5**. An oven-dried 10 mL of Schlenk tube was charged with arylpyrrole **4** (0.20 mmol), 1 mL cyclohexane and *rac*-**C8** (0.01 mmol) at ambient temperature. Then, ketomalonate **2a** (0.10 mmol) was added to the above solution and the mixture was stirred until the starting material was completely consumed. The mixture was concentrated under reduced pressure and purified by flash column chromatography (ethyl acetate/petroleum ether) to afford the corresponding racemic product **5**. Notably, there was also by-product **5′** obtained, which is the isomer of **5**.

**General procedure for the kinetic resolution of 4**. Under nitrogen atmosphere, an oven-dried 10 mL of Schlenk tube was charged with asymmetric arylpyrroles *rac*-**4** (0.40 mmol), (*S*)-**C8** (0.02 mmol), 2.4 mL of dry cyclohexane, and the mixture was stirred at 30 °C for 10 min. Then, a solution of ketomalonate **2a** (0.20 mmol) in dry cyclohexane (2.4 mL) was added dropwise to the above solution and the mixture was stirred until the starting material was completely consumed, then the mixture was concentrated under reduced pressure and purified by flash chromatography eluted with PE/EA (10/1 to 5/1) to afford the corresponding axially chiral arylpyrroles product **5** and recovered substrates (*R*)-**4**.

## Data availability

The X-ray crystallographic coordinates for structures reported in this Article have been deposited at the Cambridge Crystallographic Data Centre (CCDC), under deposition numbers CCDC 1868124 and CCDC 1868125. These data can be obtained free of charge from The Cambridge Crystallographic Data Centre via http://www.ccdc.cam.ac.uk/ data_request/cif. Supplementary information and chemical compound information are available in the online version of the paper. For NMR analysis and HPLC traces of the compounds in this article, see Supplementary Figures.  Correspondence and requests for materials should be addressed to B.T. and J.W.

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

## Acknowledgments

Financial support from the National Natural Science Foundation of China (Nos. 21772081 and 21825105), Shenzhen Nobel Prize Scientists Laboratory Project (C17213101), Shenzhen Special Funds for the Development of Biomedicine, Internet, New Energy, and New Material Industries (JCYJ20170412151701379 and KQJSCX20170328153203) is greatly appreciated.

## Author contributions

L.Z. performed experiments and prepared the supplementary information. S.-H.X. helped with characterizing some new compounds. B.T., J.W., J.-Q.W., and J.X. conceived and directed the project and L.Z., S.-H.X., J.W., and B.T. wrote the paper.

## Additional information

**Competing interests:** The authors declare no competing interests.

