## [Peer Review File · Nature Communications]

Reviewers' comments:

Reviewer #1 (Remarks to the Author):

The manuscript by Wang and Tan is an interesting study of an enantioselective alkylation of pyrroles using two different strategies of action: desymmetrization and kinetic resolution. The reaction catalyzed using a chiral phosphoric acid, allows to prepare interesting scaffolds realizing the remote control of the axial chirality with very good yields and enantioselectivity. This reaction represents a novel and complementary study to other methods reported by the same and other authors wherein the realization of the remote control of the chiral axis was achieved using the prochiral tert-butyl derivative as Michael or Friedel-Crafts acceptor and not as donor like in this case. Regarding both mechanisms reported, the scope is large and complete. Further derivatizations highlight the importance and the generality of the project presented. Furthermore these products work well also as ligands for asymmetric Pd catalyzed allylic alkylation. This aspect represents an important example since the usefulness of this kind of molecules is indeed demonstrated. I think the manuscript is appropriate for publication to Nature Communications after some major and minor revisions.

Major revisions:

The authors provide some experiments where they show the configurational stability of the enantioenriched pyrrole derivative 3a by heating it in different solvents. Because these substrates are new products, and furthermore stable atropisomers, it would be nice, in order to have a complete characterization, to have an experimental and calculated value of the corresponding energy barrier for compounds 3 and 5.

In table 3 there is no correspondence between the absolute configuration of all compounds 5 and the absolute configuration obtained from the X-ray experiment. The structure of compounds 5 accounts for a product with R_a absolute configuration and not S_a as indicated in the text and deduced from X-ray analysis. Please check carefully the absolute configuration of your products and reports the correct structures of compounds 5 in table 3 and in the SI where they are reported as S_a as showed by the X-ray analysis.

The proposed transition state is not very clear. To make the activation mode of the acid clearer a new image should be reported in my opinion. Indeed it would be nice to know if the reaction proceed in the absence of an acidic catalyst, that would support the possible interaction of the phosphate with the H₃ of the pyrrole.

Minor revisions:

In figure 2b when the project is presented, the nature of the substituents related to the two transformations realized, should be specified. For example R1 = R2 Desymmetrization, R1 ≠ R2 kinetic resolution.

Table 2: the scope of the desymmetrization is presented with a scheme of reaction where the R substituent refers to the electrophilic species, while in the table the R group in blue, refers to the substituent of the aromatic group. The same is also for R' in the scheme and R' in the table. This can create confusion. Please provide a new order of the substituents.

In reference section line 256 the journal is reported two times.

Reviewer #2 (Remarks to the Author):

This manuscript describes the atroposelective addition of an electrophilic ketone to an N-aryl pyrrole mediated by a chiral Brønsted acid. From a fundamental reactivity perspective this is not a particularly inventive approach: there are a very significant number of reactions of pyrroles with electrophiles in the presence of chiral Brønsted acids, and the recent work from Bencivenni is clearly the inspiration for this desymmetrization approach.

The major quandary for this reviewer is therefore whether the limited conceptual novelty in strategy is outweighed by the success of the strategy in its application. On balance, I believe that this is worthy of publication, and after a number of stylistic and linguistic changes feel that this could be an interesting and valuable paper.

The substrate scope is broad and relatively wide ranging, and the enantioselectivities range from excellent (where there is an ortho tert-butyl group on the N-aryl substituent), to merely good (where there is almost any other group in this position). The authors also perform a kinetic resolution, with uniformly impressive S-factors. Overall, there is a great deal of work in the paper backed up by high quality SI.

I would suggest changing the 'remote control' label in the title of the paper, which to me suggests that the catalyst or the atroposelectivity is a consequence of some external or remote stimulus (which it is not). This appears to me an homage to the Bencivenni paper and is, in my view, unnecessary. The authors also refer, in the abstract to 'optically pure axially chiral arylpyrroles'. This is a misstatement; the authors mean enantioenriched.

Reviewer #3 (Remarks to the Author):

In this paper, Tan and co-workers report on the desymmetrization or kinetic resolution of symmetrically or disymmetrically substituted aryl pyrroles, respectively. These resolutions are performed via acid catalyzed 1,2-addition of the pyrrole to ketomalonates, and good to excellent enantioselectivities were achieved by using chiral phosphoric acids as organocatalysts.

The methodology enables the introduction of a functionality at position 3 of the pyrrole ring, but this is far away from the stereogenic axis, thus limiting their utility as chiral ligands. One of the products (from a substrate that was pre-functionalized with a phosphino group) was indeed tested as a ligand in a well-known asymmetric allylic alkylation, but a previous HPLC preparative separation was required because the product was obtained with a relatively low 89% ee). Another point: the configurational stability, measured apparently to assess the suitability of the products in catalysis, was determined for the tBu-substituted product 3a, but then is 3o the product that finds application in catalysis. The configurational stability of 3o (or any other potentially useful ligands) should therefore be checked instead.

It is also claimed that the halogenated products could be used for further coupling reactions, but no examples are provided. The obtention of C-N and C-P coupling products from the ortho substituted halides would indeed enhance the synthetic value of the paper, especially if any application could be shown.

Another aspect that should be considered: the ketomalonate is used as the limiting reagent, and the more elaborated aryl pyrrole is used in excess. It would be more convenient to run experiments with the pyrrole as the limiting reagent, so that the yields would be more meaningful.

On the other hand, the products are claimed to be bioactive, but this is just a speculative assumption.

The proposed activation mode is logical but there is no experimental (NMR) nor computational support, and consequently is simply too speculative and should be removed unless additional supporting evidence is provided.

Summarizing, this is an interesting reaction with potential applications, but it lacks at this point the general interest that is required for publication in Nature Communications.

Our Responses to the Comments of Referees

Reviewer #1 (Remarks to the Author):

The manuscript by Wang and Tan is an interesting study of an enantioselective alkylation of pyrroles using two different strategies of action: desymmetrization and kinetic resolution. The reaction catalyzed using a chiral phosphoric acid, allows to prepare interesting scaffolds realizing the remote control of the axial chirality with very good yields and enantioselectivity. This reaction represents a novel and complementary study to other methods reported by the same and other authors wherein the realization of the remote control of the chiral axis was achieved using the prochiral tert-butyl derivative as Michael or Friedel-Crafts acceptor and not as donor like in this case. Regarding both mechanisms reported, the scope is large and complete. Further derivatizations highlight the importance and the generality of the project presented. Furthermore these products work well also as ligands for asymmetric Pd catalyzed allylic alkylation. This aspect represents an important example since the usefulness of this kind of molecules is indeed demonstrated. I think the manuscript is appropriate for publication to Nature Communications after some major and minor revisions.

Our response: We appreciate the positive comments of the referee.

Major revisions:

The authors provide some experiments where they show the configurational stability of the enantioenriched pyrrole derivative 3a by heating it in different solvents. Because these substrates are new products, and furthermore stable atropisomers, it

would be nice, in order to have a complete characterization, to have an experimental and calculated value of the corresponding energy barrier for compounds 3 and 5.

Our response: We thank the referee for this suggestion. Some configurational stability experiments have been performed with (*R*)-**3a**, (*R*)-**5c** as well as (*R*)-**3o** as the model and the experimental results are summarized in **Supplementary Table S6-S8**.

Table S6. Studies of the configurational stability of (*R*)-3a**.**

entry ^a	T (°C)	Time (h)	ee (%) ^b	solvent
1	25	24	95	i PrOH
2	50	24	95	
3	80	24	95	
4	100	24	95	
5	120	36	95	
6 ^c	150	12	95	
7 ^c	150	24	95	
8	80	36	95	DCE
9 ^c	100	24	95	
10 ^c	80	36	95	toluene
11 ^c	110	24	95	
12 ^c	130	24	95	
13 ^c	150	12	95	
14 ^c	150	24	95	

^aThe configurational stability of the product was studied by heating a solution of (*R*)-**3a** (0.02 mmol) in solvent (2.0 mL). ^bThe ee values were determined by HPLC analysis using a chiral stationary phase. ^cCompound **3a** was partially decomposed.

Firstly, product (*R*)-**3a** retained its high enantioselectivity (95% ee) after stirring at up to 150 °C in different solvents (*i*PrOH, DCE and toluene) for 24 hours.

Table S7. Studies of the configurational stability of (*S*)-5c****

entry ^a	T (°C)	Time (h)	ee (%) ^b	solvent
1	25	24	90	i PrOH
2	50	24	90	
3	80	24	90	
4	100	24	90	
5	120	36	90	
6 ^c	150	17	90	
7 ^e	150	30	-	
8	80	36	90	DCE
9	100	24	90	
10	110	24	90	
11	140	24	90	
12	150	30	90	
13 ^c	160	12	90	
14 ^c	160	16	90	
15 ^d	170	12	-	
16	80	36	90	toluene
17	110	24	90	
18	130	24	90	
19 ^c	150	17	90	
20 ^c	150	30	90	
21 ^d	160	12	88	
22 ^d	160	16	88	
23 ^e	170	12	-	

^a The configurational stability of the product was studied by heating a solution of (*S*)-**5c** (0.02 mmol) in solvent (2.0 mL). ^b The ee values were determined by HPLC analysis using a chiral stationary phase. ^c Compound **5c** was partially decomposed. ^d Compound **5c** was mostly decomposed. ^e Compound **5c** was totally decomposed.

Next, product (*S*)-**5c** retained its enantioselectivity (90% ee) after stirring at 150 °C in different solvents (*i*PrOH, DCE and toluene). Despite a lower 88% ee was observed by heating (*S*)-**5c** at 160 °C in toluene for 12 hours, most of (*S*)-**5c** was decomposed.

Table S8. Studies of the configurational stability of (*R*)-3o****

entry ^a	T (°C)	Time (h)	ee (%) ^b	solvent
--------	----------	---------------------	---------

1	25	24	91	iPrOH
2	50	24	91	
3	80	24	91	
4	100	24	91	
5	110	48	91	
6	120	16	91	
7	120	32	91	
8 ^c	130	12	90	
9 ^c	130	36	89	
10 ^d	140	12	89	
11 ^e	140	36	-	
12	80	36	91	DCE
13	100	24	91	
14	110	24	91	
15	110	48	91	
16 ^c	120	16	90	
17 ^d	120	32	-	
18	80	36	91	
19	100	24	91	
20	100	48	91	
21	110	48	91	
22 ^c	120	16	91	
23 ^c	120	32	90	
24 ^c	130	12	90	
25 ^d	130	36	-	

^a The configurational stability of the product was studied by heating a solution of (*R*)-**3o** (0.02 mmol) in solvent (2.0 mL). ^b The ee values were determined by HPLC analysis using a chiral stationary phase. ^c Compound **3o** was partially decomposed. ^d Compound **3o** was mostly decomposed. ^e Compound **3o** was totally decomposed.

Finally, product (*R*)-**3o** retained its enantioselectivity (91% ee) after stirring at 110 °C in different solvents (*i*PrOH, DCE and toluene) for up to 48 hours. Only 1%-2% ee erosion was observed by heating (*R*)-**3o** up to 130 °C in both *i*PrOH and toluene. At the same time, (*R*)-**3o** was started to decompose rapidly.

The above experimental results indicated that the axial chiralities of the test products (**3a**, **5c** and **3o**) are highly stable in different solvents (*i*PrOH, DCE and toluene). Deterioration of stereochemical integrity was negligible even under the conditions which the substrates began to decompose. As a consequence, it is quite difficult to calculate the value of energy barrier of these compounds by the routine methods.

In table 3 there is no correspondence between the absolute configuration of all compounds **5** and the absolute configuration obtained from the X-ray experiment. The

structure of compounds **5** accounts for a product with *R_a* absolute configuration and not *S_a* as indicated in the text and deduced from X-ray analysis. Please check carefully the absolute configuration of your products and reports the correct structures of compounds **5** in table 3 and in the SI where they are reported as *S_a* as showed by the X-ray analysis.

Our response: Thanks the referee for pointing out our mistake on drawing the chemical structure of compounds **5**.

We have carefully checked the X-ray crystal structure of compound **5a** and the correct absolute configuration should be (*aS*) according to the rules for assigning the absolute configuration at a chiral carbon atom (*Angew. Chem. Int. Ed.* **1966**, *5*, 385-415; *Angew. Chem. Int. Ed.* **1982**, *21*, 567-583; and *Einführung in die Organische Stereochemie* by Buxton S. R. and Roberts S. M.). It should be noted that the absolute configuration for compounds **5a-5l** is (*aS*) and compounds **5m-5n** is (*aR*), respectively. The detailed determination method was shown as follows and all the corrections have been made throughout the manuscript and supporting information for compound **5a-5n**.

Accordingly, the **Table 3** in the revised manuscript is presented as follows:

Table 3. Substrate scope with respect to asymmetric racemic arylpyrroles.^a

The proposed transition state is not very clear. To make the activation mode of the acid clearer a new image should be reported in my opinion. Indeed it would be nice to know if the reaction proceeds in the absence of an acidic catalyst that would support the possible interaction of the phosphate with the H3 of the pyrrole.

Our response: We would like to thank the referee for this valuable suggestion. In order to give more mechanistic insights for this transformation, a series of control experiments were then conducted without **CPA** catalyst as suggested. As depicted in the following **Scheme 1**, the reactions between **1a** and **2a** in *c*-hexane at room temperature for 24 hours gave the desired **3a** in more than 50% yields, indicating the strong background reaction for this transformation. Meanwhile, the reactions between **4a** and **2a** could also proceed in the absence of acidic catalyst, albeit lower yields. These results clearly illustrated C3 of the pyrrole is an applicable nucleophile to attack the ketomalonate. Based on the reported literatures and the above results, we anticipated that the **CPA** interacts with ketomalonate via double H-bond and enhance the electrophilicity of the ketone of ketomalonate. However, it is hard to predict the possibility of the interaction between **CPA** and H3 of the pyrrole with these initial results at this stage.

Scheme 1. Control experiments in the absence of CPA catalyst.

To test the probability of the interaction between CPA and H3 of the pyrrole, a series of NMR monitoring experiments were performed. First, a solution of (*S*)-**C8** (18.24 mg, 1.20 equiv) and **1a** (4.54 mg, 0.02 mmol) was stirred in CDCl₃ (1.0 mL) at rt for 12 hour. As depicted in **Figure 1**, identical chemical shifts were observed (5.87 ppm) for H3 and H4 of pyrrole even in the presence of excess CPA (1.20 equiv).

Figure 1. ¹H NMR monitoring experiments with **1a** and CPA in CDCl₃.

Meanwhile, a variation of the chemical shift (-0.14 ppm) was recorded as compared to the ^{31}P NMR of CPA as shown in **Figure 2**.

Figure 2. ^{31}P NMR monitoring experiments with **1a** and CPA in CDCl_3 .

Subsequently, the NMR monitoring experiments were performed in *c*-hexane- d_{12} (the developed reaction solvent is *c*-hexane) and identical chemical shift was detected too.

Figure 3. ^1H NMR monitoring experiments with **1a** and CPA in *c*-hexane- d_{12} .

Similarly, a variation of the chemical shift (+0.20 ppm) was recorded as compared to the ^{31}P NMR of CPA as shown in **Figure 4** with *c*-hexane- d_{12} as the solvent.

Figure 4. ^{31}P NMR monitoring experiments with **1a** and **CPA** in *c*-hexane- d_{12} .

As shown in **Figures 1-4**, no variation of the chemical shift was detected for H3 and H4 of the pyrrole by the ^1H NMR spectra analysis in both CDCl_3 and *c*-hexane- d_{12} even in the presence of excess **CPA**. On the other hand, inconspicuous variations of the chemical shift (-0.14 and +0.20 ppm) were observed for the **CPA** by the ^{31}P NMR spectra analysis with both CDCl_3 and *c*-hexane- d_{12} as the solvent. The above results demonstrated the possible weak interaction between **CPA** and *N* atom of the pyrrole and then ruled out the possibility of the interaction between **CPA** and H3.

Finally, to verify the interaction between **CPA** and the other substrate ketomalonate, a solution (*S*)-**C8** (7.6 mg, 10 mol%) and **2a** (0.10 mmol) was stirred in CDCl_3 (1.0 mL) at rt for 12 hours. A variation of the chemical shift (-0.36 ppm) was recorded by ^{31}P NMR spectra analysis (**Figure 5**).

Figure 5. ^{31}P NMR monitoring experiments with **2a** and **CPA** in CDCl_3 .

Meanwhile, -0.61 ppm of the chemical shift was observed in *c*-hexane- d_{12} as shown in

Figure 6.

Figure 6. ^{31}P NMR monitoring experiments with **2a** and **CPA** in *c*-hexane- d_{12} .

Overall, the more obvious variations of the chemical shifts clearly demonstrated that the interaction between **CPA** and ketomalonate should be much stronger than **CPA** and pyrrole (-0.14 vs -0.36 ppm in CDCl_3 and 0.2 vs -0.61 ppm in *c*-hexane- d_{12}). The observed identical chemical shift for H3 of pyrrole in both CDCl_3 and *c*-hexane- d_{12} ruled out the interaction between **CPA** and H3 at the beginning of this reaction.

Then a more reasonable catalytic cycle for this transformation was proposed based on the above experimental results and the reported literatures (**Scheme 2**).

Scheme 2. Plausible mechanism for this transformation.

Minor revisions:

In figure 2b when the project is presented, the nature of the substituents related to the two transformations realized should be specified. For example $R^1 = R^2$ Desymmetrization, $R^1 \neq R^2$ kinetic resolution.

Our response: We have specified the two transformations according to the suggestion of the referee and the revised Figure 2b is presented as follows:

Table 2: the scope of the desymmetrization is presented with a scheme of reaction where the R substituent refers to the electrophilic species, while in the table the R group in blue, refers to the substituent of the aromatic group. The same is also for R' in the scheme and R' in the table. This can create confusion. Please provide a new order of the substituents.

Our response: We very much thank the referee for raising this issue. As suggested, we have re-ordered the substituents of the substrates and uniformed the colors used in the chemical structures to avoid the confusion. Accordingly, the revised Table 2 is presented as follows:

Table 2. Substrate scope with respect to symmetric prochiral arylpyrroles.^a

^a Unless otherwise stated, all reactions were carried out with **1** (0.30 mmol), **2** (0.20 mmol), (*S*)-**C8** (5 mol%) in *c*-hexane (3.0 mL) at room temperature. ^b Gram-scale reactions with **1a** (8.7 mmol) and **2a** (5.8 mmol). ^c Conducted with **1** (0.20 mmol), **2a** (0.60 mmol), (*S*)-**C8** (10 mol%) in *c*-hexane/methylcyclohexane (1.5 mL/1.5 mL) at -30 °C. ^d **1** (0.24 mmol) was used. ^e Conducted at 30 °C. ^f Conducted at 50 °C.

In reference section line 256 the journal is reported two times.

Our response: We are sorry for the mistake and we have made the correction in the revised manuscript.

Reviewer #2 (Remarks to the Author):

This manuscript describes the atroposelective addition of an electrophilic ketone to an N-aryl pyrrole mediated by a chiral Brønsted acid. From a fundamental reactivity perspective this is not a particularly inventive approach: there are a very significant number of reactions of pyrroles with electrophiles in the presence of chiral Brønsted acids, and the recent work from Bencivenni is clearly the inspiration for this desymmetrization approach. The major quandary for this reviewer is therefore whether the limited conceptual novelty in strategy is outweighed by the success of the strategy in its application. On balance, I believe that this is worthy of publication, and after a number of stylistic and linguistic changes feel that this could be an interesting and valuable paper.

Our response: We deeply appreciate the positive comments of the referee and we have removed the following stylistic and linguistic issues raised by the referee.

The substrate scope is broad and relatively wide ranging, and the enantioselectivities range from excellent (where there is an ortho tert-butyl group on the N-aryl substituent), to merely good (where there is almost any other group in this position). The authors also perform a kinetic resolution, with uniformly impressive S-factors. Overall, there is a great deal of work in the paper backed up by high quality SI.

Our response: It was great appreciated for these positive comments and suggestions of the referee.

I would suggest changing the ‘remote control’ label in the title of the paper, which to me suggests that the catalyst or the atroposelectivity is a consequence of some external or remote stimulus (which it is not). This appears to me an homage to the Bencivenni paper and is, in my view, unnecessary.

Our response: We thank the referee for this suggestion and the title of this paper has been changed to “Phosphoric Acid Catalyzed Atroposelective Construction of Axially Chiral Arylpyrroles” in the revised manuscript.

The authors also refer, in the abstract to ‘optically pure axially chiral arylpyrroles’. This is a misstatement; the authors mean enantioenriched.

Our response: In accordance with the suggestion of the referee, related misstatement “optically pure axially chiral arylpyrroles” has been corrected to “enantioenriched axially chiral arylpyrroles” in the abstract.

Reviewer #3 (Remarks to the Author):

In this paper, Tan and co-workers report on the desymmetrization or kinetic resolution of symmetrically or disymmetrically substituted aryl pyrroles, respectively. These resolutions are performed via acid catalyzed 1,2-addition of the pyrrole to ketomalonates, and good to excellent enantioselectivities were achieved by using chiral phosphoric acids as organocatalysts. The methodology enables the introduction of a functionality at position 3 of the pyrrole ring, but this is far away from the stereogenic axis, thus limiting their utility as chiral ligands.

One of the products (from a substrate that was pre-functionalized with a phosphino group) was indeed tested as a ligand in a well-known asymmetric allylic alkylation, but a previous HPLC preparative separation was required because the product was obtained with a relatively low 89% ee).

Our response: We thank the referee for the valuable suggestions. As a relatively low 89% ee was obtained for the significant product **3o** under the standard conditions, we then tried to improve the enantioselectivity by modifying the optimized conditions. Gratifyingly, the ee value could be increased to 91% when the reaction was conducted in 15 °C within 60 hours. Besides, the reaction of compound **1o** and **2b** in *c*-hexane with (*S*)-**C8** (5 mol%) as the catalyst at 30 °C within 96 hours could give the desired product **3u** as a solid in 80% yield and 91% ee. It should be noted that the ee value of compound **3u** could be enhanced to 99% by recrystallization (Et₂O/hexane) instead of chiral preparative HPLC separation. Furthermore, this compound has been identified as a chiral ligand in asymmetric catalysis. All these results are summarized as follows and the above information has been included in revised manuscript.

Scheme 3. The improvement of the ee for **3o** and the synthesis and application of **3u**

Another point: the configurational stability, measured apparently to assess the suitability of the products in catalysis, was determined for the tBu-substituted product 3a, but then is 3o the product that finds application in catalysis. The configurational stability of **3o** (or any other potentially useful ligands) should therefore be checked instead.

Our response: We would like to thank the referee for this valuable suggestion. The configurational stability of compound **3o** has been examined with different solvents in different temperatures and the results are summarized in Table S8 in the revised SI.

Table S8. Studies of the configurational stability of (*R*)-3o**.**

entry ^a	T (°C)	Time (h)	ee (%) ^b	solvent
1	25	24	91	iPrOH
2	50	24	91	
3	80	24	91	
4	100	24	91	
5	110	48	91	
6	120	16	91	
7	120	32	91	
8 ^c	130	12	90	
9 ^c	130	36	89	
10 ^d	140	12	89	
11 ^e	140	36	-	
12	80	36	91	DCE
13	100	24	91	
14	110	24	91	
15	110	48	91	
12 ^c	120	16	90	
13 ^d	120	32	-	
16	80	36	91	toluene
17	100	24	91	
18	100	48	91	
19	110	48	91	
20 ^c	120	16	91	
21 ^c	120	32	90	
22 ^c	130	12	90	
23 ^d	130	36	-	

^a The configurational stability of the product was studied by heating a solution of (*R*)-**3o** (0.02

mmol) in solvent (2.0 mL).^b The ee values were determined by HPLC analysis using a chiral stationary phase. ^c Compound **3o** was partially decomposed. ^d Compound **3o** was mostly decomposed. ^e Compound **3o** was totally decomposed.

It is also claimed that the halogenated products could be used for further coupling reactions, but no examples are provided. The obtention of C-N and C-P coupling products from the ortho substituted halides would indeed enhance the synthetic value of the paper, especially if any application could be shown.

Our response: We have conducted the following coupling reactions with compound **3h** as the starting material as suggested by the referee. As summarized in **Scheme 4**, the Sonogashira reaction proceeded smoothly to furnish the desired product **16** with a synthetic useful alkynyl group in 82% yield, while compound **17** was produced efficiently in 63% yield by Suzuki coupling with organoboronic acid in the presence of palladium catalyst. Apart from that, the verified axially chiral phosphine ligand **3o** could be synthesized in 66% yield via C-P bond formation. Notably, no ee erosion was detected for all these reactions. We have included these results in the revised manuscript. Despite the extension to C-N bond formations was attempted with many classic approaches (for instance: Buchwald-Hartwig coupling, Ullmann coupling), no satisfactory result was obtained till now.

Scheme 4. The coupling reactions with **3h** as the starting material

Another aspect that should be considered: the ketomalonate is used as the limiting reagent, and the more elaborated aryl pyrrole is used in excess. It would be more convenient to run experiments with the pyrrole as the limiting reagent, so that the yields would be more meaningful.

Our response: We are grateful for this suggestion. Indeed, the aryl pyrrole substrates prepared by classic Paal-Knorr reactions are more elaborated and the experiments with pyrrole as the limiting reagent are more meaningful. However, the chemical yields and the enantioselectivities were obviously decreased when the reaction were

performed using pyrrole as the limiting reagent from a series of control experiments. The detailed experimental results are summarized in **Scheme 5**. As a consequence, we prefer to utilize ketomalonate as the limiting reagent for a better reaction outcome for this work.

Scheme 5. Control experiments with ketomalonate or pyrrole as the limiting reagent.

On the other hand, the products are claimed to be bioactive, but this is just a speculative assumption.

Our response: Thank the referee for pointing out this issue for us. We have corrected the inappropriate statement “and also an important precursor for further synthetic transformations into bioactive compounds” to “and also an important precursor for further synthetic transformations into highly functionalized pyrroles with potential bioactivity” in the abstract.

The proposed activation mode is logical but there is no experimental (NMR) or computational support, and consequently is simply too speculative and should be removed unless additional supporting evidence is provided.

Our response: We would like to thank the referee for the valuable suggestion. We have conducted a series of control experiments and NMR monitoring experiments to check the proposed activation mode. As the results shown in **Schemes 1-2** and **Figures 1-6** of this response letter, the more obvious variations of the chemical shift demonstrated that the interaction between **CPA** and ketomalonate should be much stronger than **CPA** and pyrrole (-0.14 vs -0.36 ppm in CDCl₃ and 0.2 vs -0.61 ppm in *c*-hexane-d₁₂). Furthermore, the observed identical chemical shift for H3 of pyrrole in both CDCl₃ and *c*-hexane-d₁₂ solvent ruled out the interaction between **CPA** and H3 of pyrrole at the beginning of the reaction. Then a more reasonable catalytic cycle were proposed for this transformation based on the above results and literatures (also see **Scheme 2**).

Summarizing, this is an interesting reaction with potential applications, but it lacks at this point the general interest that is required for publication in Nature Communications.

Our response: We would like to thank the referee for these valuable comments and suggestions, which are of great importance for us to improve our research work. First a wide spectrum of synthetic transformations were performed with enantioenriched **3a** as the starting material to test the practicality of the obtained product. Subsequently, the classic Sonogashira coupling and Suzuki coupling were realized by employing **3h** with an iodo substituent as the reagent. In addition, the C-P bond could be formed to give product **3o** possessing a phosphino group (**Figure 3** in the manuscript).

Furthermore, to expand the application in asymmetric catalysis, we test the possibility of the axially chiral arylpyrrole with a phosphino group as the ligand for palladium catalyzed asymmetric allylic alkylation. Fortunately, the reaction of racemic **18** and malonate **19** proceeded effectively with 2 mol% of palladium catalyst and 4 mol% of **3o** to give the desired product **20** in 95% yield with 97% ee. Aside from **3o**, compound **3u** (99% ee, after recrystallization) could also work well in this type of reaction with indole as the nucleophile, indicating that the resulted enantioenriched axially chiral arylpyrrole is capable of inducing the chirality in asymmetric synthesis. These results are summarized in **Figure 4a** in the revised manuscript.

Despite the deep and systemic exhibition of the general applications in asymmetric catalysis as well as other related domains of the produced axially chiral arylpyrroles is quite difficult at current stage, the representative transformations and applications have demonstrated the potential applications in synthetic and catalytic chemistry. It is expected that more and more applications will emerge with the establishment of the asymmetric catalytic approach for the construction of highly enantioenriched axially chiral arylpyrrole scaffold.

REVIEWERS' COMMENTS:

Reviewer #1 (Remarks to the Author):

The revised version of the manuscript can now be accepted for publication since all the major and minor revisions have been successfully addressed.

Reviewer #3 (Remarks to the Author):

The paper has been substantially improved after considering the concerns raised by me, as well as the other reviewers. I think the quality of this paper is now higher and I can recommend publication in Nat. Comm.

Point-by-point response to the referees

We would like to thank the referees for their positive comments about our work. We would also like to thank them for their kind help and valuable suggestions on improving our work.

REVIEWERS' COMMENTS:

Reviewer #1 (Remarks to the Author):

The revised version of the manuscript can now be accepted for publication since all the major and minor revisions have been successfully addressed.

Our response: We deeply appreciate the positive comments of the referee.

Reviewer #3 (Remarks to the Author):

The paper has been substantially improved after considering the concerns raised by me, as well as the other reviewers. I think the que quality of this paper is now higher and I can recommend publication in Nat. Comm.

Our response: We deeply appreciate the positive comments of the referee.